# Barriers and enablers to the implementation of immediate postpartum and post-abortion family planning service integration in primary health care units of Wolaita Zone, Southern Ethiopia: A baseline study for implementation research

**Mengistu Meskele**[1]*, **Fekadu Elias Sadamo**[1], **Banchialem Nega Angore**[2], **Samson Kastro Dake**[1], **Wondwosen Mekonnen**[3], **Addisalem Titiyos Kebede**[3], **Yohannes Mihretie Adinew**[3], **Bilal Shikur**[4], **Meselech Assegid**[4], **Naod Firdu**[4], **Senait Seid**[4], **Abiy Seifu**[4]

**1** Department of Reproductive Health and Nutrition, School of Public Health, College of Health Sciences and Medicine, Wolaita Sodo University, Wolaita Sodo, Ethiopia, **2** School of Midwifery, College of Health Sciences and Medicine, Wolaita Sodo University, Wolaita Sodo, Ethiopia, **3** Engenderhealth Ethiopia, Addis Ababa, Ethiopia, **4** School of Public Health, College of Health Science, Addis Ababa University, Addis Ababa, Ethiopia

* mengistu77@gmail.com

## Abstract

### Introduction

Evidence indicates that postpartum and post-abortion women accept family planning at a higher rate when offered timely at appropriate sites. Therefore, this study explored barriers and enablers of postpartum and post-abortion family planning utilization in primary health care units of Wolaita Zone, Southern Ethiopia, from June 20 to July 25, 2022.

### Methods

We used a case study strategy of qualitative research using both the Consolidated Framework for Implementation Research (CFIR) and Gender, Youth, and Social Inclusion (GYSI) frameworks was conducted from June to July 2022. We conducted 41 in-depth and key informant interviews and six focus group discussions. We also used Open Code software version 4.02 for coding and further analysis and applied a framework analysis.

### Results

The analysis of this study identified barriers and enablers of postpartum and post-abortion family planning service uptake in five CFIR domains and four GYSI components. The barriers included misconceptions and sole decision-making by husbands, cultural and religious barriers, and healthcare providers paying less attention to adolescents and husbands, which prevented them from using immediate postpartum and postabortion family planning services. The health facilities were not adequately staffed; there was a shortage and delay

**Data Availability Statement:** All relevant data are within the paper and its Supporting information files.

**Funding:** We obtained grant from Engender health Ethiopia: Grant award Number: CFP073 Engender Health Agreement Number: PSET002/2022 N.B: The funders had no role in study design, data collection and analysis, decision to publish, or preparation of the manuscript.

**Competing interests:** The authors have declared that no competing interests exist.

of supplies and infrastructure, trained staff turnover, and poor accountability among service providers. The existence of community structure, equal access and legal rights to the service, and having waivered services were enablers for postpartum and post-abortion family planning service uptake.

## Conclusion and recommendation

The current study identified various barriers and enablers to the uptake of postpartum and post-abortion family planning. Therefore, there is a need for high-impact interventions such as targeting male partners and girls, ensuring infrastructure, supplies, and equipment, building staff capacity, and making decisions jointly.

## Introduction

Globally, among the 1.9 billion women in the reproductive age group in 2019, 1.1 billion need family planning (FP), and 270 million have an unmet need for contraception [1]. The overall prevalence of unmet need for FP among married women and women of reproductive age groups is 22.4% and 16.2%, respectively [2, 3]. Moreover, the unmet need for FP among married women in the Damot Woyde district of Wolaita was 26.3% [4]. Providing integrated services is one of FP's promising high-impact practices (HIPs). Integrating services can save clients time and money, minimize pressures on the healthcare system, and decrease individual physician workloads [5]. Postpartum and post-abortion care service points are the critical contact points to integrate FP services. Evidence indicates that women who have received post-abortion services accept FP at a higher rate when offered at the same time and location as post-abortion care. There is also evidence that FP information provided as part of antenatal care in the third trimester, delivery, and postpartum period has a positive association with postpartum contraceptive use [6, 7].

Family planning has been integrated into human Immunodeficiency Virus (HIV) services, and it was reported that facilities that integrated FP-HIV were found to increase contraceptive use and decrease the unmet need for contraception. The health system factors constraining integration include human resource turnover and shortages, lack of policy guidance on integrated care, poor insight, unclear service delivery guidelines, inadequate infrastructure, and inadequate monitoring systems [8]. Evidence suggests that FP interventions contributed to a reduction in the maternal mortality ratio [9, 10]. The national family planning guideline of Ethiopia also recommends post-abortion family planning (PAFP) and immediate postpartum family planning (IPPFP) counseling and services to be provided before being discharged from the facility [10]. A study in Ethiopia reported the integration of IPPFP counseling into postnatal care services to be an effective means to increase postpartum contraceptive uptake [7]. Evidence on integrating family planning with postpartum,post-abortion, and other health services remains weak [11]. Thus, this indicated the need for well-designed evaluation research. The literature also showed that no comprehensive study has been conducted examining the integration of family planning services with health services [11]. Studies in Ethiopia on postpartum family planning utilization were observational and lacked interventional studies, or implementation study designs to provide evidence-based interventions to improve postpartum family planning uptake [12].

Therefore, this study aimed to explore barriers and enablers of postpartum and post-abortion family planning utilization in primary health care units of Wolaita Zone, Southern Ethiopia.

## Methods and materials

### Study setting

We conducted this formative assessment in the catchment of 15 health centres in six selected Wolaita Zone, Southern Ethiopia districts. Wolaita Zone is one of the zones in the South Ethiopia Region. It is structured into 16 districts and six town administrations; the administrative centre of the Zone is Wolaita Sodo town. Based on the 2021 population projection by the Central Statistical Agency (CSA), the total population of the Zone is estimated to be 6,142,063; of these, females count 3,115,050, and males count 3,027,013 [13]. One comprehensive and referral hospital, six public hospitals, four private hospitals, 68 health centres, and other private health facilities deliver family planning services to the population in the Zone [14].

### Study approach and period

We used a case study design using various data-gathering techniques [15]. We have used the Gender, Youth, and Social Inclusion (GYSI) and Consolidated Framework for Implementation Research (CFIR) framework tools to capture rich qualitative data, analyze, and report actionable findings on sociocultural, contextual, and intervention-related factors. The GYSI includes four dimensions: Asset and Resources, Practice, roles and participation, Knowledge, belief and perception, and Legal rights and status. The CFIR comprises five domains and 39 constructs. The domains include intervention characteristics, inner and outer settings, characteristics and attitudes of individuals, and implementation process. The study was conducted, and participants were recruited from June 20 to July 25, 2022.

### Sample size and study participants

We conducted a total of 41 in-depth/key informant interviews, of which ten were conducted with recently delivered women, ten with husbands of recently delivered women, eight with health centre heads, eight with maternal and child health (MCH) focal persons, two with woreda maternal health experts, and one with regional health bureau MCH expert. Moreover, three FGDs with boys and three with girls were conducted. The FGD participants were in the age group of 10–19 years. We recruited study participants from project implementation health facilities and respective catchment communities. A purposive sampling procedure was used to identify recently delivered women and husbands, adolescents of both sexes, healthcare providers and leaders such as health centre heads, maternal and child health (MCH) focal, and district and regional experts. The sample size was determined using the principle of "saturation". Women, husbands, healthcare providers and leaders were asked to participate in interviews until additional interviews did not provide evidence about the main themes of interest [16].

### Data collection

We adopted a semi-structured interview guide from GYSI tools and used the CIFR domains/construct [17] during a consultative workshop that has been further used for data collection (S1 File). We adopted separate interview guides to interview different categories of participants from June 20/2022 to July 25, 2022. We pilot-tested the interview guide and made refinements to make sure the guide could elicit the information we intended to capture. The piloting was conducted among healthcare providers and leaders who were not included in this study. Based

on the pilot interviews, we paraphrased some of the GYSI questions. Also, we added more probing questions to get detailed information on the different aspects of integrating IPPFP and IPAFP interventions. The interview guides were translated into the local language by experts in the two languages.

Five senior research assistants (RAs) who hold Master's degrees and have rich experience in qualitative research collected the data. We provided two days of training for the RAs on participant selection and the interview guides. We conducted the interviews face-to-face in a quiet place and ensured privacy to enable participants to feel free while expressing their opinions. The interviews were 50 to 125 minutes long. All interviews were audio recorded and transcribed verbatim and then translated into English. The same RAs transcribed and translated all the recorded interviews. Finally, we exported the data to Open Code software version 4.02 for coding and further analysis.

## Trustworthiness

Based on the following standards, we upheld the data collection's scientific rigour and reliability [18]. Credibility: The primary investigator spent time in the field and gathered data (S2 File) from the study participants to ensure the study accurately represents the participants' opinions. We included the participant's verbatim quotes in the results section, and following each interview, we had regular debriefing sessions with crucial research team members. Transferability: We have included in-depth descriptions of the study participants. Dependability: The research team audited the data daily with the supervisor and co-supervisor.

Furthermore, we shared the data with respondents to ensure it accurately reflected their experiences. Confirmability: We used the audit trail, where the principal researcher kept meticulous procedural records of the study process and reflexivity. The former demonstrates what the principal researcher knows about the participants and himself.

## Data analysis

We applied a framework analysis approach using CFIR domains and GYSI components. First, we did multiple reviews of the transcripts and tape records to familiarize ourselves with the data. Second, we coded using appropriate phrases described below; third, we assigned codes using the CFIR domains and constructs. Fourth, we generated code reports from Open code to reflect GYSI dimensions. Fifth, we developed analytic summaries using the CFIR construct and GYSI dimensions. Finally, we determined whether the construct/domains negatively or positively influenced the implementation of IPPFP and PAFP services.

The core research team, including the PI, did the coding. We initially coded the first five transcripts of varied participants. We then independently coded the transcripts and assigned appropriate CFIR constructs. We initially considered all 39 CFIR constructs and their definitions to assign codes to capture factors that might influence the implementation of IPPFP and PAFP. These CFIR codes were analytical because they required the coder to interpret the data and then apply the CFIR code that reflected a potential barrier or facilitator being described.

We were cautious in applying the fewest codes possible to make the data more accessible for analysis. When coding the data, we made four decisions. First, we assigned the appropriate operational code (e.g., room shortage). Second, we identified which of the five CFIR domains reflected the principal implementation theme in the data (e.g., Outer setting). Third, we determined which CFIR code within that identified domain was reflected in the data segment and assigned the appropriate contextual code (e.g. Patient needs and resources). In the third step, we applied codes to capture the principal implementation theme in the data segment by applying only one CFIR code per CFIR domain. Our general rule was to use one CIFR domain per

data segment. However, we made exceptions if two CFIR domains were equally reflected in the data segment. During such a case, we selected both domains and applied a CFIR code for each but did not apply more than one CFIR code per domain. This helped us avoid over-applying codes by focusing our interpretation on the most relevant CFIR constructs in the data. We used one code for each data segment during analysis, even if two or more constructs were used during the initial coding. Overall, we used 13 CFIR constructs to analyze and report our data. Finally, at the fourth stage, we indicated an'√' sign for constructs/domains reflected in the GYSI dimensions (e.g. Asset and resources) and an 'X' sign for those not.

## Ethical statement

We obtained the Ethical clearance from the Research Ethical Committee of the School of Public Health, Addis Ababa University (S3 File). In addition, we got permission from local government offices. Written informed consent and permission for those boys and girls below 18 years were obtained from the participants and parents/guardians, respectively, after the researcher had explained the purpose, procedures, benefits, and risks of the study. We anonymized any information that could identify the individual participants during or after data collection. We also respected the respondent's right to refuse some or all questions. In addition, we kept the privacy and confidentiality of participants' information. All audio records were stored in a password-protected computer that can be accessed by the investigators only. We finally disseminated the findings to health extension workers, local health leaders, and the community. We also presented the paper at the 35th conference of the Ethiopian Public Health Association (EPHA), and we will publish it in an internationally reputable journal.

## Results

We conducted 41 in-depth and key informant interviews and six FGDs.We have included the details of sociodemographic characteristics of our participants in the table below (Table 1).

**Table 1. Socio-demographic characteristics of the study participants in Wolaita Zone, Southern Ethiopia; July 2022.**

| Variable | | Frequency | Percentage |
|---|---|---|---|
| Age (years) | <18 | 16 | 17.6 |
| | 18–25 | 49 | 53.8 |
| | 26–35 | 28 | 30.8 |
| | 35+ | 8 | 8.8 |
| Sex | Male | 63 | 69.2 |
| | Female | 28 | 30.8 |
| Educational status | Not attended formal education | 4 | 4.4 |
| | Primary education | 16 | 17.6 |
| | Secondary education | 46 | 50.5 |
| | College and above | 25 | 25.5 |
| Occupation status | Government employee | 26 | 28.6 |
| | Student | 47 | 51.6 |
| | Housewife | 7 | 7.7 |
| | Farmer | 5 | 5.5 |
| | Other* | 6 | 6.6 |

*Daily labour, Merchant

## Domains of CFIR and GYSI components

The findings from this formative assessment reported CFIR domains and GYSI components using illustrative quotations. In addition to our narrative report with CFIR domains, we presented the linkage between CFIR domains and GYSI components with a summary table (S4 File).

## Domain 1: Intervention characteristics

In this domain, three constructs- relative advantage, complexity, intervention design quality, and packaging- emerged as important factors influencing the implementation of IPPFP and PAFP interventions.

**Relative advantage.** Our findings regarding service providers' and communities' perceptions of the advantages of implementing IPPFP and PAFP were varied. The clients believe that using natural methods of FP is better than using modern FP immediately after giving birth or having an abortion.

**One of the participant women said:**

*". . . Women will not have sexual intercourse until they bring their child to church for blessing. They may have sexual intercourse after 45 days of delivery even though they live in the same house."*

On the contrary, health workers strongly believe that IPPFP and PAFP services need to be strengthened, and it is a lifesaving intervention to address maternal and child mortality and minimize the unmet need for contraceptives.

**One of the HCWs reported**: *". . .when we look into the significance of postpartum and post-abortion family planning, it is essential. When a woman uses it, she keeps her health and can achieve her other life goals. It reduces maternal and child deaths and supports their mental health very well.*

**Complexity.** Mothers perceive that it is challenging to use FP immediately after giving birth and abortion. They believe women get tired and are in pain immediately after giving birth. They also perceive that women will face health risks and side effects if they use FP services while not having a balanced diet.

**One of the women reported as:** *". . .Women think that taking long-acting family planning, especially IUD, can create pain and discomfort before their uterus returns. So, they fear using it before 45 days of delivery."*

**Intervention design quality and packaging.** According to our findings, it has been suggested that information should be provided to all stakeholders, such as healthcare providers, health extension workers, the community, and adolescents.

**One of the HCWS reported:** *"It is better if all health workers understand the service equally. The health care workers, the community, and the health extension workers have little information about immediate postpartum and post-abortion family planning services."*

*". . .nothing is done for adolescents in our community. Healthcare workers don't address adolescents. They are neglected groups."*

## Domain 2: Outer-setting

**The patient's needs and resources.** Health centres offer 24-hour services, with most FP services accessible nearby. Clients are satisfied with their choice, and youth-friendly services (YFS) are also utilized. However, health posts are often closed, making access difficult.

*"...in all health centres, health workers serve the community 24 hours a day... anyone can get services from a health centre at any time."*

Girls usually give birth immediately after marriage, and male partners prefer to have many children. On the other hand, girls like to get abortion services from private facilities.

*"A young girl usually goes to another place to get abortion services. We have a trained professional on CAC, but they still go to other places to get the service."*

The study participants have reported that some clients from remote villages suffer from getting transportation and cannot get the services. "*It costs about 60 birrs to come to the health centre by motorcycle from Charicho kebele. People can use motorcycles only if they do not have access to other means of transportation. Charicho is difficult to access by motorcycle, and people come to (the health centre) on foot. They cannot even get motorcycle access."*

**External policy and incentives.** Most healthcare workers, women, and husbands reported that FP is legally allowed in the immediate postpartum period.

*"Yeah, contraceptives are legally allowed during the postpartum or post-abortion period. But there should be a woman's agreement to take contraceptive methods."*

They also reported that young women can choose and use FP information and methods. Most of the participants believe that it is legally allowed to use safe abortion services. However, some of them think that it is a sin.

*"...those who come for abortion service from rural areas resist using contraceptives when we advise them after they have received abortion service.....many women perceive it as sin..."*

## Domain 3: Characteristics and attitudes of HCWs and community members

**Individual stage of change.** The findings of this study revealed that clients' acceptance of FP, delivery, safe abortion, and post-abortion services is increasing over time. *"...women perceived us as evil when we counsel them about contraceptives immediately after birth. But it's changing; they are getting familiar with immediate postpartum family planning services."*

*Self-efficacy*

Our study suggests that decisions regarding the use of healthcare, including postpartum family planning services, should be made jointly with the spouse.

"*It's better if she uses it after discussing it with her husband. Both partners should decide* it."

**Knowledge and beliefs.**   Clients use IPPFP and PAFP, with primiparous women often hesitant due to parental homestays and fear of side effects or religious prohibition.

*"I didn't want to use family planning immediately after birth. . . I refused because I didn't want to use it immediately. I wanted to stay and return to use the service."*

Misconceptions about PAFP, contraceptive side effects, Implanon's potential to cause abortion, dietary habits, and fertility return after birth have led to women's reluctance to use contraceptives.

*". . .those who want to use it (FP), there should be something like milk and sorghum in the house so that she can take care of herself. She should be able to get some fluids to drink because it is an injection that causes a burning sensation."*

## Domain 4: Inner setting and context

**Availability of resources.**   Participants reported regular supply of FP supplies from the Ethiopian Pharmaceutical Supply Agency and partner organizations, sharing excess commodities with nearby health centres and having transportation facilities.

*"We have all options of contraceptives. We get it monthly from Engender and EPSA directly by requesting through RRF."*

Health centres use their income to buy supplies, and YFS service is available at health posts for youth. Moreover, they reported that health facilities are accessible to people.*"Health centers use their budget from the income they generate in the health care financing system. The health centre finances itself in this way."*

On the other hand, most participants reported needing supplies and resources (condoms at school, FP methods at health posts, budget, and making health facilities accessible to the communities). Furthermore, participants explained that there was a delay in supply refill, old delivery coaches, Shortage of rooms, transportation problems to go too far districts, issues with road access, absence of enough ambulances, and lack of electricity and water. They reported that infrastructures like rooms are small and crowded.

*". . .we need to have separate delivery, ANC, and family planning rooms, but now we have merged the family planning room and ANC room due to a shortage of rooms."*

*Access to knowledge and information.* Healthcare workers reported conducting regular conferences for pregnant women (PWCs) to improve service utilization.

*". . .to improve mothers' use of postpartum and post-abortion family planning, we have to strengthen the pregnant women's conference, which is being held every month, and educate them about family planning."*

Moreover, they provide counselling on the pros and cons of each contraceptive method, about IPPFP during ANC & PNC, and about giving birth at a health facility.

*". . .now, we are counselling mothers starting from their ANC visit, and we provide family planning after birth and before leaving the health facility to return to their home."*

The study participants also explained that they have trained health workers on IPPFP & PAFP. Trained Integrated Emergency Surgery and Obstetrics (IESO) provide the services. The trained midwives in hospitals need refresher training.

*"We have trained health workers. After training, the trained health workers oriented other health workers on-job."*

Community sources of information on IPPFP and health-related issues include health professionals, HEWs, HDA leaders, mass media, classmates, friends, mothers, YFS, family members, and school clubs.

*"They (Adolescents) get information from the health extension workers, health care workers, and from the radio."*

**Networks and communications.**    Health workers regularly meet with HDA, 1 to 5 networks, staff, and pregnant women, relying on Kebele women's affairs for support and collaboration with other stakeholders and NGOs.

*"...we have different forms of team among the community like health development army and 1 to 5 networks. So, these groups support each other. We have close contact with health development army leaders. We inform them, and they disseminate information to others under their supervision."*

**Readiness for implementation.**    Poor commitment of healthcare workers, poor performance, mishandling of equipment, Shortage of trained staff, and staff turnover contribute to poor commitment and weak performance in remote areas.

*"...we have a shortage of trained professionals in safe abortion care. Often, trained health workers shift to other facilities like hospitals and health centres near urban areas."*

**Leadership engagement.**    Participants also described cultural and religious leaders as barriers to using IPPFP and PAFP. They reported that some religious leaders don't allow women to use FP.

*"Some clients, specifically youth not married yet, do not want to use family planning services here. That is because we know each other. So, they fear rejection by the community since the community culturally does not allow sexual relations before marriage."*

*"Some people support family planning, and others do not. Some people say it is against the will of God to use it, especially the wives of religious leaders who don't want to use it."*

## Domain 5: Process

**Planning, engaging, and leadership.**    Health facilities varied in preparing plans for IPPFP and PAFP, which need improvement and monitoring.

*"I am not seeing promising changes regarding this service. We do this based on emotion, not critically thinking and planning the activities. Because of this, we are not observing changes at the grassroots level."*

In addition, model women engaged in providing information to the community.
*". . ..during the pregnant women's conference, we have to use women who received the service as role models."* There were varied perceptions towards the uptake of FP methods and women's leadership. Recently, women have been engaged in leadership positions in the MCH department and church leadership.

*". . .nowadays women are assuming leadership positions better than before. They are participating very well at government offices or religious institutions."*

Executing, reflecting, and evaluating.Healthcare workers are willing to reasonably provide IPPFP and PAFP services to everyone in all facilities. A mechanism is in place to monitor and evaluate the performance of IPPFP and PAFP services.

*"I work on checking the proper registration of documents and make sure the necessary equipment and materials are ready to support the coming delivery, such as gloves, Normal Saline, and other materials."*

## Discussion

This study aimed to explore barriers and enablers of immediate postpartum and post-abortion family planning utilization in primary health care units. This study identified various barriers and enablers of the utilization of immediate postpartum and post-abortion family planning and its service uptake. This study also identified that IPPFP and PAFP utilization among adolescents was inadequate. There was also a negative perception among some providers toward adolescent girls' use of IPPFP and PAFP. This result is consistent with previous study results in Ethiopia, Malawi, and Nigeria, where IPPFP utilization was low over the years, including for childbearing adolescents [19]. This could be due to some barriers to using PAC services, including a lack of proper access, poor care, and an unfriendly setting for adolescents [20].

Additionally, a previous study suggests that the reasons for not using Immediate Postpartum family planning were breastfeeding, fear of side effects, husband/partner opposition and infrequent sex [21]. Therefore, studies conducted in Sub-Saharan Africa suggest some strategies that improve post-abortion services, like misoprostol and manual vacuum aspiration (MVA), are equally secure and efficient. Moreover, the previous studies highlight that physicians and skilled mid-level cadres are similarly successful in PAC administration. PAC should also be administered right before a patient leaves the facility, and then PAC contraceptive uptake rises [20].

Furthermore, in this study, some care providers didn't encourage unmarried youth and those from rural areas to use family planning and mistreated them. Participants reported that nothing is done for adolescents in our community. Healthcare workers didn't address adolescents' IPPFP and PAFP issues. Moreover, participants reported that adolescents were neglected groups in such services.

Our study identified family planning commodities unavailability, stock out, and poor stock management systems in some health facilities that negatively affected IPPFP and PAFP service uptake. Earlier study results support the current result in that ensuring the supply of family

planning methods increases the uptake of IPPFP [22]. The previous study confirmed the present study's finding that when family planning services are unavailable in contact with the health system, clients were forced to refer and could not be provided with the method as soon as possible [19, 23, 24]. However, earlier studies confirmed the results of the current study, finding that most health facilities had family planning methods and there were no stockouts [25]. Participants reported the resource availability due to the regular supply of family planning supplies by EPSA, Engenderhealth, and Amref. Moreover, the participants suggested their strategy to avoid Shortage as those health centres that owned excess family planning methods had to be shared with nearby health centres with a shortage of ways to exchange a technique with them to solve the Shortage. Earlier study results confirmed our result [22], which is collaboration with partners like non-governmental organizations in family planning commodities.

This study revealed that there are different misconceptions widely distributed within the community. According to the current research, some community members believe Implanon can cause abortion. In contrast, others think it migrates to other body parts, and some believe that contraceptive use is associated with dietary habits. Previous study findings from Washington and Nepal confirmed that family planning misperceptions were barriers to IPPFP use [26, 27]. Another similar conclusion from the Philippines and elsewhere reported that cultural and social factors play a significant role in family planning [4, 25].

Moreover, women who live in rural areas believe that using contraceptives after abortion is a sin. Similarly, participants in the community believe that fertility will not return until two years after giving birth, and husband refusal deterred the IPPFP service uptake. Therefore, men's involvement in the postpartum family planning consultation and awareness creation in each family planning method is highly encouraged [28–30].

Our study has reported that a food shortage at the household level and being from a low-income group affects family planning utilization. A previous study conducted in North West Ethiopia also confirmed that low income affects post-abortion service utilization [31]. A prior study in Ethiopia showed similar findings to the current research in which women who did not receive delivery care from healthcare professionals had lower odds of receiving IPPFP [32].

The findings of this study revealed that male partners oppose the uptake of immediate postpartum and post-abortion family planning utilization. Study findings of Ouagadougou, Burkina Faso, also showed husbands' refusal to refrain from unprotected sex was a challenge [28]. Another past study result also confirms this finding in which the reluctance of men to attend antenatal clinics (ANC) was a barrier to IPPFP use [33]. On the other hand, previous studies also revealed male partners enhance IPPFP and PAFP utilization [30, 34].

Women perceived in this study that it is challenging to use family planning immediately after giving birth and abortion. It was also reported that women get tired and are in pain immediately after birth. Moreover, culturally, women in our community believe that pregnancy is unexpected because they attend religious ceremonies, and some first-time mothers stay with their parents for three months postpartum, delaying the timing of return to sexual activity. This is in line with the findings from Western Ethiopia [29, 30]. Therefore, there is a need to counsel family planning to increase the utilization of IPPFP and PAFP [35, 36].

Study participants also explained the poor commitment of healthcare workers at some health facilities as they are unwilling to teach in remote areas. However, previous study findings suggest that providing counselling on IPPFP and PAFP leads to increased uptake of postpartum and post-abortion family planning [36, 37]. Moreover, in the current study, participants reported that HEW performance is occasionally weakening. This poor commitment was exacerbated as government employees were not getting their full salary at the right

time, but only half/quarter of it. This is supported by another study where receiving allowance was correlated with the satisfaction of the health workers and contraceptive prevalence rate [38]. Participants also revealed in this study that the mishandling of equipment leads them to be non-functional. Shortage of trained staff [30] and their turnover, and training provided to specific health care providers, for instance, only for midwives, leads to the service complex. Therefore, our findings suggest additional and appropriate training needs for healthcare providers on family planning [22].

Our finding echoed an earlier study by Save the Children that suggests the importance of regular monthly PWCs [39]. Participants also reported that they were conducting regular PWCs to improve service utilization. Moreover, they forwarded the action as they counselled about each method's pros and cons, counselled about IPPFP during ANC & PNC, and counselled women to give birth at health facilities. Previous studies show that PWCs are also very effective in tackling harmful traditional norms and practices and improving mothers' care-seeking behaviour and newborns' care [39].

## Strength and limitations

This study was based on identifying information from various study participants, both data and method triangulation, using the standard CIFIR and GYSI tools as a strength. The involvement of multiple professionals from three universities, Engender Health, conducting various review meetings, and making necessary corrections after the discussions with all research teams were also strengths. However, the study findings are difficult to generalize to other contexts. Moreover, the research may be influenced by the researcher's preferences and personal beliefs, though the study tried to bracket our ideas aside. Persuading readers who are used to precise statistical solutions might be challenging. Excluding recently delivered adolescents and those who have had abortions from the study may limit the overall understanding of IPPFP and PAFP needs among all sexually active adolescents, as these groups may have unique perspectives and experiences that could inform future interventions. Additionally, finding ways to include these groups in research while respecting their privacy concerns should be a priority to ensure a comprehensive understanding of adolescent IPPFP and PAFP needs.

Additionally, while self-reported data can provide valuable insights, it may not always accurately reflect a facility's true conditions or practices. Conducting audits of the facilities would have provided more objective and concrete evidence of the barriers and enablers of IPPFP and PAFP use.

## Conclusion and recommendation

The study highlighted barriers to integrating IPPFP and PAFP, including providers' mistreatment, misconceptions, supplies and resource shortage, poor staff motivation, household food shortage and low income, partner opposition, and poor commitment. Still, it also suggested availability, collaboration, and networking, conducting PWCs, and having regular meetings as facilitators.

Therefore, based on the findings, we recommend that FP methods-related supplies and resources be available at all health facilities. Awareness should be created among young girls and women on FP and CAC since they have many misconceptions. Moreover, religious leaders should be taught that male partners should be involved in IPPFP and PAFP and jointly make decisions. Training healthcare staff and fulfilling their benefits will increase the staff's motivation. Household income generation activities need to be promoted to improve household food security. Making health facilities accessible to each district is also strongly recommended.

## Supporting information

**S1 File. Questionnaire _English and Amharic version.**
(DOCX)

**S2 File. Data_transcripts_merged.**
(PDF)

**S3 File.**
(PDF)

**S4 File. Formative assessment analysis based on CFIR domains and GYSI components.**
(DOCX)

## Author Contributions

**Conceptualization:** Mengistu Meskele, Fekadu Elias Sadamo, Addisalem Titiyos Kebede, Yohannes Mihretie Adinew, Abiy Seifu.

**Data curation:** Mengistu Meskele, Fekadu Elias Sadamo, Banchialem Nega Angore, Samson Kastro Dake, Addisalem Titiyos Kebede, Meselech Assegid, Naod Firdu, Senait Seid, Abiy Seifu.

**Formal analysis:** Mengistu Meskele, Fekadu Elias Sadamo, Banchialem Nega Angore, Samson Kastro Dake, Addisalem Titiyos Kebede, Yohannes Mihretie Adinew, Meselech Assegid, Senait Seid, Abiy Seifu.

**Funding acquisition:** Mengistu Meskele, Addisalem Titiyos Kebede, Abiy Seifu.

**Investigation:** Fekadu Elias Sadamo, Banchialem Nega Angore, Samson Kastro Dake, Wondwosen Mekonnen.

**Methodology:** Mengistu Meskele, Fekadu Elias Sadamo, Banchialem Nega Angore, Samson Kastro Dake, Wondwosen Mekonnen, Addisalem Titiyos Kebede, Yohannes Mihretie Adinew, Bilal Shikur, Meselech Assegid, Naod Firdu, Senait Seid, Abiy Seifu.

**Project administration:** Mengistu Meskele, Fekadu Elias Sadamo, Yohannes Mihretie Adinew, Senait Seid, Abiy Seifu.

**Resources:** Fekadu Elias Sadamo, Wondwosen Mekonnen, Addisalem Titiyos Kebede, Yohannes Mihretie Adinew, Bilal Shikur.

**Software:** Mengistu Meskele, Fekadu Elias Sadamo, Samson Kastro Dake, Meselech Assegid.

**Supervision:** Mengistu Meskele, Fekadu Elias Sadamo, Samson Kastro Dake, Wondwosen Mekonnen, Addisalem Titiyos Kebede, Yohannes Mihretie Adinew, Bilal Shikur, Naod Firdu, Senait Seid, Abiy Seifu.

**Validation:** Mengistu Meskele, Banchialem Nega Angore, Wondwosen Mekonnen, Addisalem Titiyos Kebede, Yohannes Mihretie Adinew, Bilal Shikur, Meselech Assegid, Naod Firdu, Senait Seid, Abiy Seifu.

**Visualization:** Wondwosen Mekonnen, Addisalem Titiyos Kebede, Yohannes Mihretie Adinew, Bilal Shikur, Naod Firdu, Senait Seid, Abiy Seifu.

**Writing – original draft:** Mengistu Meskele, Fekadu Elias Sadamo, Banchialem Nega Angore, Samson Kastro Dake.

**Writing – review & editing:** Mengistu Meskele.

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
