## [Decision Letter · Decision Letter 0]

31 Aug 2023

PONE-D-23-12106Barriers and enablers to the implementation of immediate postpartum and post abortion family planning service integration in Primary Health Care Units of Wolaita Zone, Southern Ethiopia: A Consolidated Framework for Implementation ResearchPLOS ONE

Dear Dr. Meskele,

Thank you for submitting your manuscript to PLOS ONE. After careful consideration, we feel that it has merit but does not fully meet PLOS ONE’s publication criteria as it currently stands. Therefore, we invite you to submit a revised version of the manuscript that addresses the points raised during the review process.

We look forward to receiving your revised manuscript.

Kind regards,

Kiddus Yitbarek, MPH

Academic Editor

PLOS ONE

Journal Requirements:

5. We are unable to open your Supporting Information file Data_Transcripts.rar. Please kindly revise as necessary and re-upload.

Reviewers' comments:

Reviewer's Responses to Questions

**Comments to the Author**

1. Is the manuscript technically sound, and do the data support the conclusions?

Reviewer #1: Partly

Reviewer #2: Partly

2. Has the statistical analysis been performed appropriately and rigorously? 

Reviewer #1: N/A

Reviewer #2: Yes

3. Have the authors made all data underlying the findings in their manuscript fully available?

Reviewer #1: No

Reviewer #2: Yes

4. Is the manuscript presented in an intelligible fashion and written in standard English?

Reviewer #1: Yes

Reviewer #2: No

5. Review Comments to the Author

Reviewer #1: This is an interesting peiece of work on improving FP services among women most in need.

General comment:

Though the author attempted to apply CFIR for FP services following births or abortion in the contexts of gender, youth or social inclusion, they did not manage to address the required points.

I am more inclined to reject the paper but would rather advise them to be specific and focused in rewriting the manuscript to convey clear message.

The following key comments:

1. The post partum and abortion FP services are already available services.what specific intervention or strategy was on your mind regarding FP fo this group. Eg. Was Gender and social inclusion the strategy in your mind? For what strategy you tried to mention relative advantage, compatiblity or complexity...? Was that for gender, youth, and sociak inclusiin in the program or servicr? This is not well said.

2. Individual characterstics ignored the women/adolscents' characterstics and expectations whike it emphasized on health workerrs

3. Though the attempt to address 39 constructs of CFIR is generally fine, the effort to address all these aspects brought shallow finding for each of the five domains. Hence, important questions pertaning implemenation startegies of the services remained unanswered. E.g. to what extent the FP service is accessible for post abortion or partum when they are done at home or traditionally? Contexts of the youths? Affordablities when done in private facilities due to free of stigma...

4. The study draw conclusions about the need to mobilize the community without assessing anything about it. Implrmentiin ot intervention strategies need to be assessed before concluded. Pls remove such conclusions. You should havr explored what implemenation problems of different interventions associated to Fp service (e.g: gender involvement, social inclusion,friendliness, community engagment and mobilization.....) this comment was partly indicated by comment #1.. so your finding shouls havr picked several challnges and facilitaors about these. If possible be advised to natrow the scope of the manuscript. This section alone is one paper. It is nice to have many focused and clear papers than one extensively but narrowly reported papers.

5. Alll other domains are challenged from lack of focus

Please present this manuscript by separating key aspects as distinct papers so that you can be informative and clear. At them moment the main question of this manuscript is not clearly answered, the context of gender, youth and social inclision not addressed and the conclusion looks common sense

Reviewer #2: Review Report

Title: Barriers and enablers to the implementation of immediate postpartum and post abortion family planning service integration in Primary Health Care Units of Wolaita Zone, Southern Ethiopia: A Consolidated Framework for Implementation Research.

Version I:

Manuscript Number; PONE-D-23-12106

Review Comments

i. On the title and abstract

Is that evaluation of the already implemented service or exploring the possibility y of integration of both services?

What was the problem with the non-integrated service delivery? Where is the panel analysis before this study?

ii. The background is not strong and the problem statement is mostly missed

iii. On the methods section

• Inconsistent stud participants. On one hand you have interviewed those who gave birth and on the other hand you have interviewed youth who didn’t have history of abortion. Even, I wisely guess the questions will slightly push the youth to have abortion in the future and seek post abortion family planning care?

• Have you assessed friendliness of the service by time, place and conditions.

• The attitude and opinion local community leaders and religious leaders were not captured.

• Additionally, the saying of the women, youth and child affairs and the local administration was not captured.

• Cite the reference of the study area and the number of population and the number of hospitals and PHCU/standard

• The methods section fails to respond to the qualification of the data collector and trustworthiness which is highly essential.

iv. On the result, discussion and conclusion section

The beginning of the result section is absorbing but fails to shorten, clarify, simplify and to maintain logical flow. In addition, it didn’t address the standard way of presenting qualitative research.

The discussion section should have theoretical and practical considerations and ground level explanations without missing the reality?

Regards,

6. PLOS authors have the option to publish the peer review history of their article (what does this mean?). If published, this will include your full peer review and any attached files.

Reviewer #1: No

Reviewer #2: No

---

## [Author Response · Author response to Decision Letter 0]

12 Oct 2023

PLOS ONE: Response for the editor and Reviewers 

Manuscript ID: PONE-D-23-12106

Title: Barriers and enablers to the implementation of immediate postpartum and post abortion family planning service integration in Primary Health Care Units of Wolaita Zone, Southern Ethiopia: A Consolidated Framework for Implementation Research

PLOS ONE

Correspondent author: Mengistu Meskele 

Journal Requirements: 

We thank the editors of PLOS ONE for this critical query. We have now followed and incorporated the journal styles. 

When you resubmit, please provide the correct grant numbers for the awards you received for your study in the ‘‘Funding Information’ section

We thank the reviewer, now we have corrected this comment.

We thank the editor. Now we have included the funding information in the cover letter and kindly ask the editor to include it at online submission on my behalf.

Thank you the reviewer, we noted and included the supporting information files as per your comment. 

5. We are unable to open your Supporting Information file Data_Transcripts.rar. Please kindly revise as necessary and re-upload.

We noted the comment, thank you editor. I will merge the files and send it as PDF_merged file

Reviewers' comments:

Reviewer's Responses to Questions

Comments to the Author

1. Is the manuscript technically sound, and do the data support the conclusions?

Reviewer #1: Partly

Reviewer #2: Partly

We thank the reviewers for these important queries. No we made the conclusion in line with the finding. 

2. Has the statistical analysis been performed appropriately and rigorously?

Reviewer #1: N/A

Reviewer #2: Yes

3. Have the authors made all data underlying the findings in their manuscript fully available?

Reviewer #1: No

Reviewer #2: Yes.

We thank the reviewer/editors for this important query. We have now uploaded our data as well, and we have included all the participants' thick descriptions in our manuscript. 

4. Is the manuscript presented in an intelligible fashion and written in standard English?

Reviewer #1: Yes

Reviewer #2: No.

We thank the reviewers and editor for this critical comment. We now have edited the English as indicated by the reviewers and editors. 

5. Review Comments to the Author:

Reviewer #1: This is an interesting piece of work on improving FP services among women most in need.

General comment:

Though the author attempted to apply CFIR for FP services following births or abortion in the contexts of gender, youth or social inclusion, they did not manage to address the required points.

I am more inclined to reject the paper but would rather advise them to be specific and focused on rewriting the manuscript to convey a clear message.

Thank you for your comment. We accepted the comment, and we managed to be specific and focused on the revised manuscript. 

The following key comments:

1. The post-partum and abortion FP services are already available services.what specific intervention or strategy was on your mind regarding FP for this group. Eg. Was Gender and social inclusion the strategy in your mind? For what strategy you tried to mention relative advantage, compatibility or complexity...? Was that for gender, youth, and social inclusion in the program or service? This is not well said.

We thank the reviewers for this concern. As a specific intervention or strategy, we have tested the model for integration of immediate PPFP and PAFP service with existing ANC, Delivery and abortion services. We have used the theory of change models for each intervention strategy. We have used three intervention strategies, namely engaging relevant stakeholders, filling equipment and supply gaps and building the capacity of community and health workers. However, it is not the objective of this paper to report the detailed intervention strategies and models tested. Therefore, as a baseline study, we studied the relative advantage, compatibility or complexity for the suggested interventions with existing interventions on which our study analysis is based. We have used GYSI dimensions mainly to develop the tools and based our analysis on them, as mentioned in the method and result section of the manuscript. 

2. Individual characteristics ignored the women/adolescents' characteristics and expectations while it emphasized on health workers

We thank the reviewer for this critical concern. The inner setting and context domain of CFIR can only discuss institutional and health workers' characteristics. That means the domain only focuses on organizations' issues, not clients. On the other hand, characteristics and attitudes of individual’s domain address only clients, and we have already included them in Table 2. 

3. Though the attempt to address 39 constructs of CFIR is generally fine, the effort to address all these aspects brought shallow findings for each of the five domains. Hence, important questions pertaining implementation strategies of the services remained unanswered. E.g. to what extent the FP service is accessible for post abortion or partum when they are done at home or traditionally? Contexts of the youths? Affordability when done in private facilities due to free of stigma…...

We thank the reviewer for this important concern. Sure, but we didn’t include all 39 constructs of the CFIR domains. We have used only 13 CFIR constructs to analyze and report our data and showed them in the method section of our main manuscript. We have used the RE-AIM model to measure the overall progress of the implementation. However, it is not the objective of this paper to report the detailed implementation outcome. 

4. The study drew conclusions about the need to mobilize the community without assessing anything about it. Implementation of intervention strategies need to be assessed before concluded. Pls remove such conclusions. You should have explored what implementation problems of different interventions associated to Fp service (e.g: gender involvement, social inclusion, friendliness, community engagement and mobilization.....) this comment was partly indicated by comment #1.. so you should have picked several challenges and facilitators about these. If possible be advised to narrow the scope of the manuscript. This section alone is one paper. It is nice to have many focused and clear papers than one extensively but narrowly reported papers.

We thank the reviewers for this concern. As the reviewer #1 pointed it in comment #4, now we have removed such conclusions that need to mobilize the community in our manuscript as a specific intervention or strategy. We have tested the model for integration of immediate PPFP and PAFP service but it is not part of the objective of this paper to report the detailed intervention strategies and models tested.

5. All other domains are challenged from lack of focus

Please present this manuscript by separating key aspects as distinct papers so that you can be informative and clear. At the moment the main question of this manuscript is not clearly answered, the context of gender, youth and social inclusion not addressed and the conclusion looks like common sense. 

We thank the reviewer for this very important query. We now have tried to get our domains focused. See Table 2 in the result section. The main framework used in this paper is CFIR and is supported by GYSI. 

Reviewer #2: Review Report

Title: Barriers and enablers to the implementation of immediate postpartum and post abortion family planning service integration in Primary Health Care Units of Wolaita Zone, Southern Ethiopia: A Consolidated Framework for Implementation Research.

Version I:

Manuscript Number; PONE-D-23-12106

Review Comments

i. On the title and abstract

Is that evaluation of the already implemented service or exploring the possibility of integration of both services?

We thank the reviewer for this critical query. This paper is part of the bigger project of implementation of immediate postpartum and post-abortion family planning service integration in Primary Health Care Units. Therefore, this is a baseline study to identify potential barriers and enablers for the implementation of immediate postpartum and post-abortion family planning service integration in Primary Health Care Units. It is to explore the possibilities of integration of both services rather than evaluating already implemented interventions. We corrected the title accordingly and included the statement that explained more about these issues in the introduction section. 

What was the problem with the non-integrated service delivery? Where is the panel analysis before this study?

ii. The background is not strong and the problem statement is mostly missed.

We thank the reviewer for this important concern. We revised the background section in the revised manuscript.

iii. On the methods section

• Inconsistent stud participants. On one hand you have interviewed those who gave birth and on the other hand you have interviewed youth who didn’t have a history of abortion. Even, I wisely guess the questions will slightly push the youth to have abortion in the future and seek post abortion family planning care?

We thank the reviewer for the concern. We recruited study participants from different groups to assess the barriers and enablers for both postpartum and post abortion services. Thus, women who gave birth were included to assess the barriers and enablers for postpartum family planning service, and the adolescents were included to assess the service on the post abortion side. The interview guides used were different for these two groups. Regarding the reviewer’s fear about pushing youth to have abortion services, we also had a similar fear when we were developing the proposal. However, the main reason for selecting the youth is that it is difficult to identify women who have had abortion services in the past because of privacy issues, and youth are the ones that use abortion services based on the quantitative baseline data we have received.

• Have you assessed the friendliness of the service by time, place and conditions?

We thank the reviewer for this concern. We have assessed friendliness of the service through using the construct of the CFIR. We have noted the findings under the sub-heading of patient needs and resources in the line number 277-280 and 376-419 as access and utilization of youth friendly service. 

• The attitude and opinion of local community leaders and religious leaders were not captured

We thank the reviewer for this important query. Actually we didn’t interview religious and community leaders for their attitude and opinions. But the study participants expressed the religious and community leader’s attitude and opinions were included in the sub-heading of this paper line 455-464. 

• Additionally, the saying of the women, youth and child affairs and the local administration was not captured.

We thank the reviewer for the concern. We did not include women, youth and child affairs and local administration in the study because, based on our objective, no such issue has a link with the administrative bodies on the integration of family planning with postpartum and post abortion services. In the current abortion law, a woman must not provide evidence to get abortion services if she claims any of the eligibility criteria.

• Cite the reference of the study area and the number of population and the number of hospitals and PHCU/standard

We thank the reviewer for pointing this out. We cited the reference in the revised manuscript. 

• The methods section fails to respond to the qualification of the data collector and trustworthiness which is highly essential.

We thank the reviewer for pointing this out. We have included the qualification of the data collectors and trustworthiness in the revised manuscript. 

iv. On the result, discussion and conclusion section

♣ The beginning of the result section is absorbing but fails to shorten, clarify, simplify and to maintain logical flow. In addition, it didn’t address the standard way of presenting qualitative research.

We thank the reviewer for this critical comment. We thoroughly revised the result section based on your comment in the revised manuscript.

♣ The discussion section should have theoretical and practical considerations and ground level explanations without missing the reality?

We thank the reviewers for these important issues. We now have included it in the discussion. 

Kindly regards,

---

## [Decision Letter · Decision Letter 1]

12 Feb 2024

PONE-D-23-12106R1Barriers and enablers to the implementation of immediate postpartum and post-abortion family planning service integration in Primary Health Care Units of Wolaita Zone, Southern Ethiopia: A Baseline Study for Implementation ResearchPLOS ONE

Dear Dr. Meskele,

Thank you for submitting your manuscript to PLOS ONE. After careful consideration, we feel that it has merit but does not fully meet PLOS ONE’s publication criteria as it currently stands. Therefore, we invite you to submit a revised version of the manuscript that addresses the points raised during the review process.

Dear Author We appreciate working with the revision of your manuscript. Reviewers of your work have raised key concerns for revision and response. Please give special emphasis on the followng points when you revise the manuscript: - Try to convey a focused thik message based a well defined framework - Improve the write-up of the manuscript in a way that it deliver the intended message in a clear and academic way - Copy edit the manuscript for any language issuesPlease read the reviewers comments at the bottom of this email or your author's page carefully and use them to improve your work. 

We look forward to receiving your revised manuscript.

Kind regards,

Kiddus Yitbarek, MPH

Academic Editor

PLOS ONE

Reviewers' comments:

Reviewer's Responses to Questions

**Comments to the Author**

1. If the authors have adequately addressed your comments raised in a previous round of review and you feel that this manuscript is now acceptable for publication, you may indicate that here to bypass the “Comments to the Author” section, enter your conflict of interest statement in the “Confidential to Editor” section, and submit your "Accept" recommendation.

Reviewer #2: All comments have been addressed

Reviewer #3: (No Response)

Reviewer #4: (No Response)

2. Is the manuscript technically sound, and do the data support the conclusions?

Reviewer #2: Partly

Reviewer #3: Yes

Reviewer #4: Yes

3. Has the statistical analysis been performed appropriately and rigorously? 

Reviewer #2: N/A

Reviewer #3: N/A

Reviewer #4: Yes

4. Have the authors made all data underlying the findings in their manuscript fully available?

Reviewer #2: Yes

Reviewer #3: Yes

Reviewer #4: Yes

5. Is the manuscript presented in an intelligible fashion and written in standard English?

Reviewer #2: No

Reviewer #3: Yes

Reviewer #4: Yes

6. Review Comments to the Author

Reviewer #2: Review Report

Barriers and enablers to the implementation of immediate postpartum and post_abortion family planning service integration in Primary Health Care Units of Wolaita Zone, Southern Ethiopia: A Baseline Study for Implementation Research.

Review Comments

I. General Comments

Consult again for the requirement of the journal E.g. Is that only conclusion or Conclusion and recommendation in the abstract section. Again see the Introduction for the same concern.

Try to revisit the whole manuscript for clarification, since it is very crucial because it is lacking in the manuscript E.g. if you read this sentence “less attention being given to adolescents and husbands that 38 hinder uptake of immediate postpartum and postabortion family planning”. Who gave less attention? Attention towards what?

BE consistent throughout the document and maintain its cope.

Language, grammar and editorial issue and study period

• Clarity E.g., in the abstract section “…a higher rate when offered at the same time and location.” can be re-written as “…a higher rate when offered timely at appropriate site”. Secondly, in the same section “aimed to explore” can be re-written as “explored”. Again, in the result section of the result section you can avoid ‘significant’ from barriers. Also, again “service free of charge” can be re-written as well “waivered services” …etc.

• Plural e.g., Conclusion

• Full stops are lacking

• The end or the introduction section have no study period.

• The formatting is incorrect

• Hence try to revisit or consult for editorial service before the next submission.

II. Specific Comments

a. On the Introduction Section

It is not concise, occupied wide space and the adverse consequences and benefits the whether use of PP and PA-family planning should be also clearly depicted.

Reference 19 is not correct and try t use Guba (1972).

Don’t tell us the science of trustworthiness, but report what you have already done to ensure quality.

You can avoid coding and put it under data analysis.

Incorporate report or plan for dissemination of the finding to the study are local admin and the community.

These methods still need enrichment E.g. How did you handle emerging issues and quality of the data collectors? How did you sample? What type of Qualitative research is it?

b. Result and the Consequent Sections

• Lacks clarity, concise and needs major refinement.

• If I were I will depict the summery of the findings in one domain and then describe it in detail in the consequent sentences.

• Stick to the main findings and avoid un-necessary findings.

• Try to slightly address the agreement and the difference between the two times of rendering service/ and frameworks findings.

• The conclusion is not conclusion.

• Put the abbreviation and acronyms separately at its due place.

• Avoid the use of ‘we” from the discussion section.

• The space of the result and the discussion section should revisit/

• Compare with the comparable one

• Have no recommendation

• Revisit the consequent section.

Regards,

Reviewer #3: Comments

Line 91 to 92: it says that “Studies thus far revealed that integrating family planning with other health services was still weak and indicated the need for well-designed evaluation research.”, But, it lack references.

Line 102 to 104: Most previously conducted studies in Ethiopia on postpartum family planning utilization were observational studies and lacked interventional study or implementation study designs to provide evidence-based interventions to improve postpartum family planning uptake (17). How many studies you reviewed for this paragraph?

Line 137 to 138: You said that “Moreover, three FGDs with boys and three with girls were conducted.” Since your study is about postpartum and post-abortion family planning service, how adolescents were your target population? I need your justification how you select your target population?

Under your sample size section, would you elaborate how you determine the sample size?

Line 142 to 143: you sad that “We adopted a semi-structured interview guide from GYSI tools and used the CIFR domains/construct “you directly adopted interview guide. Do you mean you did not modify something, please?, why you do a consultative workshop? How you contextualize the tool?

Line 162 to 164: Credibility: The primary investigator spent time in the field and gathered data from the healthcare providers to make sure the study accurately represents the opinions of the participants. Please check it, is it healthcare providers?

Under Trustworthiness section, lacks member checking? Why?, If you do it add it? If not why?

Result section lacks who says what?

Under Discussion section, why you discuss the finding based on domain?

Line 500, our study has reported a household food shortage and low income in the community? What do you mean food shortage?

Line 587: An earlier study confirms our finding and suggests the importance of conducting PWC every month. What does this means, would you make it clear, please?

Please add the limitation and strength of the study?

Reviewer #4: (No Response)

7. PLOS authors have the option to publish the peer review history of their article (what does this mean?). If published, this will include your full peer review and any attached files.

Reviewer #2: No

Reviewer #3: No

Reviewer #4: No

---

## [Author Response · Author response to Decision Letter 1]

28 Mar 2024

We thank the reviewer and editor for this concern. Now we have corrected the heading as conclusion and recommendation as both are in the abstract section together. Similarly, the introduction is now also corrected. 

We thank the reviewer for this concern. Now we have corrected the ambiguity as “healthcare providers paying less attention to adolescents and husbands, which prevented them from using immediate postpartum and postabortion family planning services”. Moreover, we have corrected the language and grammer as indicated by the reviewers.

We accepted the reviewer's suggestions for this concern and we have now addressed the queries raised in the introduction and result section of the abstract.

We thank the reviewer and now we have included the study period. we have corrected the language and grammar as indicated by the reviewers.

We thank the reviewer and now we made the introduction concise and clear. 

We thank the reviewer and now we replaced reference number 19 with the correct one. Guba,

Trustworthiness reference was included as indicated by the reviewer. We included only what we have done and removed the science about trustworthiness.

We thank the reviewer for this concern. Now we deleted the sub-heading “coding” and put it under data analysis as per your suggestion. 

We thank the reviewer. Now we have included the dissemination plan under the section of Ethical consideration as “We finally disseminated the findings to health extension workers, local health leaders, and the community. We also presented the paper at the 35th conference of the Ethiopian Public Health Association (EPHA) and we will publish the paper in an internationally reputable journal” 

We thank the reviewers. We have corrected as indicated by the reviewers “We recruited study participants from project implementation health facilities and respective catchment communities. A purposive sampling procedure was used to identify recently delivered women and husbands, adolescents of both sexes, healthcare providers and leaders such as health center heads, maternal and child health (MCH) focal, and district and regional experts. The sample size was determined using the principle of “saturation”— women, husbands healthcare providers, and leaders were asked to participate in interviews until additional interviews did not provide additional evidence about the main themes of interest. We used a case study design using various data-gathering techniques’’. 

We thank the reviewer for this important query. We deduced the redundant parts and made them concise as indicated by the reviewers. Moreover, 

we have removed a very lengthy CFIR and GYSI table and included it as supplementary files. 

We thank the reviewer and now we made the conclusion concise as indicated. 

We thank the reviewer for this suggestion and we now have moved the abbreviations and acronyms at their due place in the manuscript. 

We thank the reviewers and no we replaced “we” by ‘’this study’’. Also. We included recommendations and amended the result and discussion sections. 

We thank the reviewer for these important points. We now removed Line number 91 and 92. 

We thank the reviewer, now we have corrected the sentences of 102 to 104 as we reviewed one article for that particular paragraph. We corrected it as “ A previous study conducted…….” 

We thank the reviewer for this concern; we just to know the abortion-related experience of adolescents. However, we encountered difficulty in identifying adolescents who had experienced abortion and we considered it as a limitation of our study. We elaborated on the sample size and included it in the method sections. 

We adopted and contextualized the tool in our co-design workshop. 

We thank the reviewer for this concern. The healthcare providers were also part of this study and now we corrected this section as “The primary investigator spent time in the field and gathered data from the study participants”

We thank the reviewer regarding the trustworthiness and now we corrected it as “ Furthermore, we shared the data with respondents to ensure it accurately reflected their experiences.” We provided the initial data that we transcribed to the respondent to cross-check whether it is consistent with their original (e.g. we checked it during our review meeting event that includes the participants like health care workers, women, husbands partners) 

We thank the reviewer for this important concern. Now we make it clear “Our study has reported that a food shortage at household level and being from a low-income group affects family planning utilization”. participants believe that they need to have an adequate and balanced diet to use FP and those from low-income households have less chance to use it. 

We thank the reviewer for this query; based on the reviewer's suggestion we have corrected the clarity issue as “ Our finding echoed an earlier study conducted by Save the Children that suggests the importance of conducting regular monthly PWCs. It can promote peer support amongst women to motivate each other to seek appropriate ANC, delivery at a health facility, PNC, and family planning” .

We thank the reviewer for this important queries. We have now included the strength and limitations of the study as “This study was based on identifying information from various study participants, data and method triangulation, and also the study used the standard CIFIR and GYSI tools as a strength. The involvement of various professionals from three universities, the involvement of Engender Health, conducting various review meetings, and making necessary corrections after the discussions with all research teams were strengths. However, the study findings are difficult to generalize to other contexts. Moreover, the research may be influenced by the researcher's preferences and personal beliefs, though we tried to bracket our ideas aside. Persuading readers who are used to precise statistical solutions might be challenging.” Excluding recently delivered adolescents and those who have had abortions from the study may limit the overall understanding of IPPFP and PAFP needs among all sexually active adolescents, as these groups may have unique perspectives and experiences that could inform future interventions. Additionally, finding ways to include these groups in research while respecting their privacy concerns should be a priority to ensure a comprehensive understanding of adolescent IPPFP and PAFP needs.

---

## [Editor Report · Decision Letter 2]

1 May 2024

Barriers and enablers to the implementation of immediate postpartum and post-abortion family planning service integration in Primary Health Care Units of Wolaita Zone, Southern Ethiopia: A Baseline Study for Implementation Research

PONE-D-23-12106R2

Dear Dr. Meskele,

We’re pleased to inform you that your manuscript has been judged scientifically suitable for publication and will be formally accepted for publication once it meets all outstanding technical requirements.

Kind regards,

Kiddus Yitbarek, MPH

Academic Editor

PLOS ONE

Additional Editor Comments (optional):

Dear Authors,

Thank you for your efforts to improve the quality of your paper. The technical aspects are now in good shape. To prepare the manuscript for publication, we recommend a thorough copy edit to ensure clear and concise language throughout. This will involve polishing the writing style, grammar, and overall flow of the paper.
---

## [Editor Report · Acceptance letter]

26 May 2024

PONE-D-23-12106R2 

PLOS ONE

Dear Dr. Meskele, 

I'm pleased to inform you that your manuscript has been deemed suitable for publication in PLOS ONE. Congratulations! Your manuscript is now being handed over to our production team.

Kind regards, 

on behalf of

Mr. Kiddus Yitbarek 

Academic Editor

PLOS ONE